# Ceragenin-Coated Non-Spherical Gold Nanoparticles as Novel Candidacidal Agents

**DOI:** 10.3390/pharmaceutics13111940

**Published:** 2021-11-16

**Authors:** Karol Skłodowski, Sylwia Joanna Chmielewska, Joanna Depciuch, Piotr Deptuła, Ewelina Piktel, Tamara Daniluk, Magdalena Zakrzewska, Michał Czarnowski, Mateusz Cieśluk, Bonita Durnaś, Magdalena Parlińska-Wojtan, Paul B. Savage, Robert Bucki

**Affiliations:** 1Department of Medical Microbiology and Nanobiomedical Engineering, Medical University of Bialystok, 15-222 Bialystok, Poland; karol.sklodowsky@gmail.com (K.S.); sylwia.chmielewska@umb.edu.pl (S.J.C.); piotr.deptula@umb.edu.pl (P.D.); ewelina.piktel@wp.pl (E.P.); tamara.daniluk@umb.edu.pl (T.D.); magdalena.zakrzew@gmail.com (M.Z.); flaps.czarnowski@gmail.com (M.C.); mticv1@gmail.com (M.C.); 2Institute of Nuclear Physics, Polish Academy of Sciences, 31-342 Krakow, Poland; joanna.depciuch@ifj.edu.pl (J.D.); magdalena.parlinska@ifj.edu.pl (M.P.-W.); 3The Faculty of Medicine and Health Sciences, Collegium Modicum of the Jan Kochanowski University in Kielce, 25-734 Kielce, Poland; bonita.durnas@ujk.edu.pl; 4Department of Chemistry and Biochemistry, Brigham Young University, Provo, UT 84602, USA; pbsavage@chem.byu.edu

**Keywords:** *Candida* spp., ceragenins, CSA-13, CSA-44, CSA-131, gold nanoparticles, nanosystems

## Abstract

Background: Infections caused by *Candida* spp. have become one of the major causes of morbidity and mortality in immunocompromised patients. Therefore, new effective fungicides are urgently needed, especially due to an escalating resistance crisis. Methods: A set of nanosystems with rod- (AuR), peanut- (AuP), and star-shaped (AuS) metal cores were synthesized. These gold nanoparticles were conjugated with ceragenins CSA-13, CSA-44, and CSA-131, and their activity was evaluated against *Candida* strains (*n* = 21) through the assessment of MICs (minimum inhibitory concentrations)/MFCs (minimum fungicidal concentrations). Moreover, in order to determine the potential for resistance development, serial passages of *Candida* cells with tested nanosystems were performed. The principal mechanism of action of Au NPs was evaluated via ROS (reactive oxygen species) generation assessment, plasma membrane permeabilization, and release of the protein content. Finally, to evaluate the potential toxicity of Au NPs, the measurement of hemoglobin release from red blood cells (RBCs) was carried out. Results: All of the tested nanosystems exerted a potent candidacidal activity, regardless of the species or susceptibility to other antifungal agents. Significantly, no resistance development after 25 passages of *Candida* cells with AuR@CSA-13, AuR@CSA-44, and AuR@CSA-131 nanosystems was observed. Moreover, the fungicidal mechanism of action of the investigated nanosystems involved the generation of ROS, damage of the fungal cell membrane, and leakage of intracellular contents. Notably, no significant RBCs hemolysis at candidacidal doses of tested nanosystems was detected. Conclusions: The results provide rationale for the development of gold nanoparticles of rod-, peanut-, and star-shaped conjugated with CSA-13, CSA-44, and CSA-131 as effective candidacidal agents.

## 1. Introduction

In recent years, the incidences of *Candida* fungal infections have substantially increased. According to various studies, the attributable mortality among all patients with candidaemia has been recorded to be between 10% and 47%; however, a more accurate estimate is approximately 10–20%, with the risk of death being strictly connected with the increasing age, the species of *Candida* strain causing the infection, the use of immunosuppressive agents, and the presence of the venous catheter, as well as pre-existing renal dysfunction and other comorbidities [1,2,3,4,5].

Systemic, potentially life-threatening infections are predominantly caused by *Candida* spp. It is worth emphasizing that nearly 10 million cases of mucosal candidiasis are reported globally and more than 150 million of people are affected by serious fungal diseases [6]. Moreover, *Candida* infections induce more than 3.6 million healthcare visits each year in the USA, resulting in $3 billion of direct medical costs [7].

It should be emphasized that *Candida albicans* is the most frequently identified species isolated in candidiasis-suffering patients. However, the role of other species such as *C. glabrata*, *C. tropicalis*, *C. parapsilosis,* and *C. krusei* have recently increased [8,9,10,11]. Most importantly, these species are more likely to be resistant to commonly used antifungal drugs and demonstrate the potential to lead to outbreaks [6]. All five of these species of *Candida* are isolated in more than 90% of invasive fungal infections [12,13]. The global burden of invasive candidiasis is approximately ~750,000 cases, and these infections cause high mortality rates, even in excess of 40% [6].

Despite the ongoing serious threat connected with candidiasis, the number of effective antimycotics available in therapy is still limited [14,15]. Additionally, growing resistance of fungi to available antimycotics poses a serious challenge to modern medicine. There is a growing number of drug-resistant fungi, including multi-drug resistant (MDR) strains on four continents, namely: Asia (Pakistan, India, and Japan), Africa (South Africa), South America (Venezuela) [16], and North America (USA/Canada) [17]. Furthermore, it is estimated that in 33,800 cases of hospitalized patients in the USA *Candida* drug-resistant species were isolated, which ultimately resulted in 1700 patient deaths in 2017. Crucially, the CDC in their last report warned against drug-resistant *Candida* and defined this pathogen as a serious threat that immediately necessitates new treatment options [7].

Antifungal drug resistance is a prominent aspect that negatively impacts the clinical outcome for patients with invasive candidiasis. It is worth noting that the most significance mechanisms of resistance to antifungal drugs involve (i) the reduction of drug intercellular accumulation through the activation of membrane-bound efflux pumps, (ii) the decrease in the affinity of agents to their targets (target mutation and target expression deregulation), and (iii) counteraction of the drug effect by metabolic modification triggered by, e.g., an echinocandins paradoxical effect or ergosterol biosynthesis pathway alteration [18,19,20].

Although advances in prophylaxis, diagnosis, and therapy have occurred, invasive *Candida* infections caused by resistant strains still contribute to significant mortality, particularly in immunocompromised patients, which underlines the urgent need to develop new antifungal drugs [6,21]. The search for newer antimycotics with improved pharmacokinetics and an enhanced ability to kill drug-resistant pathogens have led scientists to nanotechnology. It should be noted that the use of nanoparticles ameliorates the pharmacodynamic and pharmacokinetic parameters of the drug, including bioavailability, release time of the active substance, and prolongation of the pharmacological action [22,23,24,25].

Crucially, in the era of constantly increasing resistance to antibiotics and antimycotics, the use of nanomaterials is an innovative therapeutic approach to combat infections, notably those caused by resistant strains [26]. In this respect, several features of NPs make them an attractive alternative to traditional antibiotics and antimycotic agents [27]. First, a large surface area to volume ratio facilitates NP interactions with microbial membranes, exerting an antimicrobial activity even in low doses, as well as allowing for efficient surface functionalization, which optimizes the activity against preferred targets [28,29]. In addition, NPs possess the ability to penetrate microorganism cell membranes and cross barriers that are typically non-permeable for conventional agents [29,30]. The therapeutic potential of NPs is also highlighted by the improvement of pharmacokinetics of poorly water-soluble drugs and the prolongation of half-life or systemic circulation [22].

A vast range of NPs produced from diverse nanomaterials has been synthesized. However, gold nanoparticles (Au NPs) reflect an important role in nanotechnology, due to their advantageous properties including low toxicity, abilities to connect to various ligands, and antimicrobial activity against resistant strains [31,32]. Au NPs are widely proposed in the development of new therapeutic agents, with possible application in diagnostics, cancer treatment, vaccines, or drug carriers [33]. It must be emphasized that many recent investigations show the antifungal activity of silver (Au) NPs against fungi. Results obtained by Paul et al. revealed that curcumin−silver nanoparticles (C–Ag–NPs) had a substantial activity against fluconazole-resistant *Candida* species isolated from patients with HIV [34]. Similar results were reported by Khatoon et al. and Edis et al., where a considerable reduction in the growth of *C. albicans, C. tropicalis* and *C. glabrata* by Au NPs was observed [35,36]. Nonetheless, the high toxicity of Au NPs significantly reduces their therapeutic applications [37]. This is why Au NPs that exhibit lower toxicity further highlight the valuable potential of use.

Although the antibacterial activities of Au NPs have been well reported, the impacts of Au NPs on fungi are not widely described in the literature. However, it is worth underlining that our previous studies proved the potent fungicidal activity of rod-shaped Au NPs against *Candida* strains and representatives of filamentous fungi from *Aspergillus*, *Cladosporium,* and *Fusarium* spp. at concentrations that are nontoxic to the host cells [38]. In line with this research direction, we decided to investigate whether Au NPs in three different shapes i.e., rods (AuR NPs), peanuts (AuP NPs), and stars (AuS NPs), can be successfully used as antifungal agents. The reason for choosing these shapes of nanoparticles was dictated by previous investigations. Piktel et al. demonstrated the paramount antimicrobial efficacy of rod-, peanut-, and star-shaped gold nanoparticles compared with spherical-shaped nanoparticles against representative isolates of *C. albicans* fungi, Gram-negative bacteria like *E. coli* and *P. aeruginosa* species, and Gram-positive strains involving *S. aureus* [39]. Moreover, the results presented by Penders et al. and Jebali et al. also confirmed that non-spherical gold nanoparticles (flowers, stars, and cubes) revealed a far superior activity than those in a spherical-shape [40,41]. The abovementioned observations therefore constitute a promising starting point for further investigations.

In recent times, more and more results have been reported indicating that the conjugation of Au NPs with other agents have shown a superior effect against microorganisms and may become a future alternative to conventional antibiotics and fungicides in the fight against MDR strains [27]. However, it has not been studied whether Au NPs coated with other agents like ceragenins are characterized by a pronounced antifungal potency. Hence, the primary goal of our investigation was to determine whether the functionalization of Au NPs with CSAs would make them prominent candidates to combat *Candida* strains. Ceragenins are synthetic compounds designed to mimic the structure and function of endogenous antimicrobial peptides (AMPs) [42]. The molecular structure of ceragenins is based on cholic acid, to which amino groups are attached to recreate the amphiphilic morphology of AMPs [43]. The chemical character between CSAs and AMP allows for the preservation of the broad-spectrum antimicrobial activity of AMPs [44]. However, ceragenins are not peptide-based; therefore, half-lives are not restricted by the action of proteases [45,46,47], and even long term storage in solutions does not affect the antimicrobial properties of ceragenins [48]. The leading mechanism of CSA action is determined by the same events as in antimicrobial peptides, which is a direct interaction with negatively charged fungal membrane components, resulting in changes in the membrane organization of phospholipids and sudden membrane disruption [49,50,51,52]. Moreover, other indirect of CSA-mediated mechanisms, such as increases in reactive oxygen species formation, have also be recognized [37,44,53]. Additionally, ceragenin functionalization on the surface of metallic nanoparticles provides the opportunity to achieve cumulative antimicrobial effects, including the ability to overwhelm some resistance mechanisms as well as obtain lower effective drug doses and a reduction of side effects of well-known antifungal agents.

To the best of our knowledge, this is the first report comparing the activities of rod-, peanut-, and star- shaped Au NPs functionalized by ceragenins involving CSA-13, CSA-44, and CSA-131 vs. ceragenins in a free form against *Candida* strains. It is noteworthy that one of the most crucial aspects of our studies was the assessment of whether fungi can develop resistance to AuR@CSA-13, AuR@CSA-44, and AuR@CSA-131 after 25 passages. Furthermore, in order to address the potential usefulness in the clinical use, the mechanisms of action and biocompatibility of the Au NPs were evaluated. This study is a starting point for creating novel nanosystems with a potent antifungal activity and promising therapeutic properties [54].

## 2. Materials and Methods

### 2.1. Candida Strains, Media, and Growth Conditions

Two reference strains (*C. albicans* ATCC 26790 and *C. albicans* 1408) and 19 clinical isolates of fungi including, *C. albicans* (*n* = 5), *C. tropicalis* (*n* = 5), *C. krusei* (*n* = 5), and *C. glabrata* (*n* = 4), were used in our investigation. Clinical strains of *Candida* were collected from hematooncological patients of Holy Cross Cancer Center in Kielce (Kielce, Poland), and reference strains were purchased from the Polish Collection of Microorganisms, Polish Academy of Science (Wroclaw, Poland). The identification and antifungal sensitivity of *Candida* strains used in this study was determined using a VITEK^®^ 2 (bioMérieux, Marcy-l’Etoile, France) utilizing fungal cards YST and YS08. This system allows for an assessment of the antimycotic susceptibility of fungi to amphotericin B (AMB), caspofungin (CSF), fluconazole (FLU), flucytosine (FC), micafungin (MCF), and voriconazole (VOR). Fungal strains were cultured and maintained on Sabouraud Dextrose agar with chloramphenicol (Biomaxima, Lublin, Poland).

### 2.2. Antifungal Compounds

#### 2.2.1. Ceragenins

Ceragenins: CSA-13, CSA-44, and CSA-131 were synthesized as described previously [55], and subsequently were dissolved in phosphate-buffered saline (PBS, Thermo Fisher Scientific, Saint Louis, MO, USA). Prepared solutions were stored in 4 °C. Tetrachloroaurate (HAuCl_4_), sodium borohydride (NaBH_4_), silver nitrate (AgNO_3_), ascorbic acid (C_6_H_8_O_6_), 16-mercaptohexadecanoic acid (MHDA), dimethylformamide (DMF), pentafluorophenyl (PFP), N,N-diisopropylethylamine (DIPEA), and N-cyclohexyl-N′-(2-morpholinoethyl) carbodiimide methyl-p-toluenesulfonate (CMC) were purchased from Merc (Darmstadt, Germany), while cetrimonium bromide (CTAB; C_19_H_42_BrN) was acquired from Carl Roth (Karlsruhe, Germany).

#### 2.2.2. Gold Nanoparticles (Au NPs) Functionalized by CSA-13, CSA-44, and CSA-131

Star-, peanut-, and rod-shaped gold nanoparticles were synthesized using the seed-mediated method. In this method, two stages of synthesis were used. In first step, gold nanoseeds with a spherical shape were prepared. For this purpose, C_19_H_42_BrN (5 mL; 5 µM), HAuCl_4_ (5 mL of 0.5 mM), and NaBH_4_ (0.6 mL; 0.1 M) were mixed with active stirring. The reaction was stopped when the solution turned red. In the second step, C_19_H_42_BrN (5 mL; 5 µM), AgNO_3_ (0.2 mL; 0.04 M), HAuCl_4_ (5 mL; 1 mM), C_6_H_8_O_6_ (70 µL; 78 mM), and 30 µL of Au nanoseeds were mixed under active stirring. For AuP NPs, the reaction was stopped after 3 h, while for AuR NPs and AuS NPs the reaction was stopped at 30 min. For AuS NPs, 210 µL of 78 mM C_6_H_8_O_6_ was added. Nanoparticles were washed and functionalized by MHDA through overnight incubation at 4 °C. The functionalization process was verified using FT-Raman spectroscopy (Thermo Fisher Scientific, Saint Louis, MO, USA). In order to immobilize ceragenins with the Au NPs-MHDA complex, the solution consisting of PFP, DIPEA, and CMC was used to rinse Au NPs conjugated with MHDA for 30 min at 25 °C. In the next step, a solution of CSA-13, CSA-44, or CSA-131 was added to the resulting solution, which was washed with DMF and centrifuged. The incubation was continued for 30 min at 25 °C.

Finally, as a result of the synthesis, the nanosystems consisted of (i) ceragenin CSA-13, CSA-44 or CSA-131 (concentration: 2 mg/mL, which equals 0.002 M, i.e., 12.04 × 10^20^ molecules) and (ii) non-spherical nanoparticles (2.93 ng/mL, equivalent to 0.015 × 10^−6^ M, i.e., 9.03 × 10^15^ molecules). The consequence was that the ceragenin charge on the nanoparticle surface was 1.33 × 10^5^ ceragenin molecules per nanoparticle. In preparing the nanosystems, we used MHDA in excess to ensure that all ceragenin molecules adhered to the nanoparticle surface. Thus, the amount of ceragenin in the prepared nanosystem was not further calculated, and was determined by the amount of ceragenins that was used for the synthesis.

#### 2.2.3. Physicochemical Properties of AuR NPs@CSA-13, AuP NPs@CSA-13, AuS NPs@CSA-13; AuR NPs@CSA-44, AuP NPs@CSA-44 and AuS NPs@CSA-44, and AuR NPs@CSA-131, AuP NPs@CSA-131, AuS NPs@CSA-131 Nanoparticles

Using scanning transmission electron microscopy (STEM, FEI, Hillsboro, OR, USA) with a high-angle annular dark field detector (HAADF) in conventional mode, the morphology and size of the obtained Au NPs were determined and performed in an aberration-corrected FEI Titan electron microscope operating at 300 kV, fitted with a FEG cathode (Field Emission Gun, FEI, Hillsboro, OR, USA). The measurements were carried out on an aberration-corrected FEI Titan electron microscope working at 300 kV equipped with a field emission gun (FEG) cathode (FEI, Hillsboro, OR, USA). The distribution of the particle size was obtained based on STEM images by counting the mean size from 100 nanoparticles placed in the different areas of the TEM grids using TIA Software version 4.0. The zeta potential distribution was determined by the microelectrophoretic method and Smoluchowski model. Each value was obtained as an average of three separate runs of the instrument with at least 20 measurements. All of the experiments were performed in water at 25 °C. Fourier transform Raman (FT-Raman, Thermo Fisher Scientific, Waltham, MA, USA) spectroscopy was used to define the efficiency of the nanoparticle functionalization and ceragenin immobilization. For this purpose, a Nicolet NXR 9650 FT-Raman spectrometer containing Nd:YAG laser (1064 nm, Thermo Fisher Scientific, Waltham, MA, USA) and a germanium detector was used (Thermo Fisher Scientific, Waltham, MA, USA). The measurements were carried out in the range of 150 to 3700 cm^−1^ with a spectral resolution of 8 cm^−1^, 128 scans, and a 1W laser power. Raman spectra were developed using Omnic/Thermo Fisher Scientific software. However, baseline correction using rubber band methods and vector normalization were performed through OPUS software version 7.0. To obtain information about the zeta potential of the obtained nanoparticles, Zetasizer Nano Series (Malvern, Worcestershire, UK) was used.

### 2.3. Methods

#### 2.3.1. MIC/MFC Measurements

The microdilution method described in the guidelines of the European Committee on Antimicrobial Susceptibility Testing (EUCAST), version 10.0-valid from 4 February 2020, was used to determine the minimum inhibitory concentrations (MICs) of the tested agents against fungal strains suspended at populations of 2 × 10^5^ CFU (colony-forming units)/mL. The final concentrations of the tested compounds ranged from 0.1 to 51.2 µg/mL. MICs were determined visually and confirmed by spectrophotometer in an RPMI medium (Sigma-Aldrich, Saint Louis, MO, USA) supplemented with MOPS (Sigma-Aldrich, Saint Louis, MO, USA) and D-(+)-glucose (Sigma-Aldrich, Saint Louis, MO, USA) at the lowest concentration of tested agents that showed no visible microbial growth after 24 h. In turn, the minimum fungicidal concentration (MFC) was determined by plating each sample (10 µL) on Sabouraud Dextrose agar with the chloramphenicol (Biomaxima, Lublincity, Poland).

#### 2.3.2. Induction of *C. albicans* ATCC 26790 and *C. krusei* 156 Resistance to AuR NPs@CSA-13, AuR NPs@CSA-44, and AuR NPs@CSA-131

MICs of AuR NPs@CSA-13, AuR NPs@CSA-44, and AuR NPs@CSA-131 against *C. albicans* ATCC 26790 and *C. krusei* 156 were determined visually and confirmed spectrophotometrically. After 24 h of incubation, the passaging was initiated by harvesting fungal cells growing at a concentration just below the MIC and inoculating into fresh RPMI medium.

Importantly, this inoculum was subjected to another MIC assay. After an 18–24 h incubation period, cells growing in the highest concentration of the antimicrobial from the previous passage were once again harvested and assayed for the MIC. The process was repeated for 25 passages.

#### 2.3.3. ROS Generation Assessment

The generation of ROS by *C. albicans* 1408 cells upon stimulation with ceragenin and Au NPs-conjugated with ceragenins was measured using 2′,7′-dichlorofluorescin diacetate (DFCH-DA, Sigma-Aldrich, Saint Louis, MO, USA) as a fluorescent probe. Fungal cells suspended in PBS (OD_600_  =  0.5) were placed in 96-well black plates and then antifungal agents were added to each well at concentrations ranging from 1 to 10 µg/mL. DFCH-DA at 20 µM was supplied and mixed. After 1 h of incubation, the relative fluorescent signals were measured at excitation/emission = 485/530 nm using a microplate reader—Labsystem Varioscan Lux (Thermo Fisher Scientific, Waltham, MA, USA). The results obtained were compared to the 5 mM concentration of the hydrogen peroxide (Sigma Aldrich, Saint Louis, MO, USA). The fungal cell visualization was achieved by DCFH-DA staining and was performed using a fluorescence microscope (Zeiss AxioObserver.A1 Fluorescence Version Inverted Optical Microscope, JPK Instruments, Berlin, Germany).

#### 2.3.4. Membrane Permeabilization Assay

Measurement of the capacity of ceragenins and ceragenins attached to the surface of gold nanoparticles to disrupt the plasma membrane of *C. albicans* 1408 cells was performed using N-phenyl-1-napthylamine (NPN; Sigma Aldrich, Saint Louis, MO, USA) as a fluorescent probe. Fungal cells suspended in PBS (OD_600_  =  0.5) were subjected to incubation with the tested agents at concentrations ranging from 1 to 10 µg/mL. Next, NPN was added to a final concentration of 0.5 mM, and the mixture was incubated for 5 min. Increases in fluorescence intensity at 348/408 nm (excitation/emission) were measured using Labsystem Varioscan Lux (Thermo Fisher Scientific, Waltham, MA, USA). Finally, the comparison of the results with the 1 μg/mL concentration of the amphotericin B (Chemat, Los Angeles, CA, USA) was prepared.

#### 2.3.5. diSC_(3)_ Assay

The ability of the studied ceragenins and ceragenin-coated nanosystems to depolarize the fungal cell membrane was investigated using a 3,3′-dipropylthiadicarbocyanine iodide (diSC_(3)_) assay. Briefly, a *C. albicans* 1408 suspension was adjusted to OD_600_ ~ 0.5. In the next step, diSC_(3)_ (Sigma Aldrich, Saint Louis, MO, USA) was added to give a final concentration of 0.4 µM, and it was incubated for 60 min at room temperature and protected from the light until a stable fluorescence signal was obtained. Thereafter, a solution of KCl (Chempur, Piekary Śląskie, Poland) (100 mM) in PBS was added to align the intra- and extra-cellular potassium concentrations, followed by 5 min of incubation. Finally, the fungal cells in suspension were incubated with studied nanosystems at concentrations of 1–10 µg/mL for 1 h. The final fluorescence value was recorded after incubation lasting 1 h using a microplate reader at 622/670 nm (excitation/emission)—Varioskan Lux (Thermo Fisher Scientific, Waltham, MA, USA). In the last step, the obtained results were compared to a 1 μg/mL concentration of the amphotericin B (Chemat, Los Angeles, CA, USA).

#### 2.3.6. Assessment of Protein Leakage

Protein leakage analysis was performed using the Bradford assay. *C. albicans* 1408 cells were resuspended in PBS with adjustment to OD_600_ ~ 0.5 prior to exposure to nanoparticles at concentrations of 1, 2, 5, and 10 µg/mL for 1 h. At the end of incubation, samples were centrifuged (5000 rpm) for 10 min. Supernatant was mixed with the Bradford reagent, i.e., Coomassie Brilliant Blue G-250 (Sigma-Aldrich, Saint Louis, MO, USA) in a 1:1 ratio. Mixtures were incubated in the dark for 10 min. The absorbance of the samples was measured at 595 nm using Labsystem Varioscan Lux (Thermo Fisher Scientific, Waltham, MA, USA). Finally, a comparison of the results with the 1 μg/mL concentration of the amphotericin B (Chemat, Los Angeles, CA, USA) was prepared.

#### 2.3.7. Haemolytic Activity Assessment

The haemolytic activity of both free and ceragenins attached to the surface of gold nanoparticles was determined with a suspension of human RBC (red blood cells) in PBS (hematocrit ~ 5%) collected from the healthy volunteers. Compounds in a concentration range of 1–50 μg/mL were incubated with RBCs for 1, 6, and 12 h at 37 °C. Upon incubation, plates were centrifuged (2500 rpm, 10 min) and the optical density of the sample was measured at 595 nm using Labsystem Varioscan Lux (Thermo Fisher Scientific, Waltham, MA, USA). A supernatant from the samples treated with 1% Triton X-100 (Sigma-Aldrich, Saint Louis, MO, USA) was used as a positive control (100% hemolysis), whereas RBCs in PBS were used as a negative control (0% hemolysis). The relative absorbance compared with that treated with 1% Triton X-100 was defined as the percentage of haemolysis.

#### 2.3.8. Statistical Analysis

All of the statistical analyses were performed sing Graph Pad Prism, version 8 (San Diego, CA, USA). The results were presented as mean ± standard deviation (SD) consisting of three to six replicate experiments. The two-tailed Student’s test was used to determine the significance of differences. A *p*-value < 0.05 was considered statistically significant.

## 3. Results

### 3.1. Physicochemical Nature of Rod-, Peanut-, and Star-Shaped Au NPs

Au NPs with different shapes and sizes of the Au NPs, depending on the synthetic methods used, were prepared. Nanoparticles with rod-like (Figure 1(A1,A2)), peanut-like (Figure 1(B1,B2)), and star-like (Figure 1(C1,C2)) shapes were obtained. However, 20% of non-rod-shaped nanoparticles and 40% of non-peanut-shaped nanoparticles were recognized. Additionally, among the rod-shape AuR NPs, rounded ends were observed along with spherical Au NPs. Sizes, measured on the longitudinal and transverse axes in AuP NPs and AuR NPs, were used to compare NPs. In the case of AuR NPs, the sizes of the axes were 45 ± 8 nm and 10 ± 3, respectively, while, in the case of the AuP NPs, the sizes were 55–65 nm and 24–34 nm, respectively. The size of AuS NPs was 243 nm. Zeta potential plots of AuR NPs, AuP NPs, and AuS NPs (Figure 1D) showed that all of the gold nanoparticles were positively charged in the entire range of pH. For AuR NPs, the zeta potential values were 49 mV for pH 3.5 to 18 mV for pH 12.5, while for AuP NPs, these values were between 42 mV and 19 mV, and for AuS NPs, the zeta potentials were from 33 mV to 16 mV. For all of these nanoparticles, the potential value decreased as the pH value increased.

Successful functionalization and immobilization of ceragenins on NPs was verified using FT Raman spectroscopy. In Figure 1E, which is a representative spectrum, the corresponding Au-S stretching vibration peaks are observed at 278 cm^−1^, which are the bonds in charge of forming the bond between the gold nanoparticles and the sulfur surface with MHDA [56]. Furthermore, successful ceragenins immobilization in the surface of the AuR NPs, AuP NPs, and AuS NPs was confirmed by the presence of peaks at 1680 cm^−1^, corresponding to the N-H vibrations created between COOH groups from MHDA and NH_2_ groups from CSA-13, CSA-44, and CSA-131, respectively [57].

### 3.2. Susceptibility of Tested Candida Strains to Antimycotics and Developed Nanoparticles

MICs and MFCs are used to determine the therapeutic potential of the antimicrobial agents. In this study, almost half of the *Candida* isolates were resistant to fluconazole. In addition, multiple strains showed intermediate resistance to caspofungin (*n* = 8) and voriconazole (*n* = 2) (according to EUCAST interpretation; Table 1). MICs and MFCs of ceragenin-appended NPs are presented in Figure 2A–C. Activities of ceragenin-based nanosystems (AuR NPs@CSA-13, AuR NPs@CSA-44, and AuR NPs@CSA-131; AuP NPs@CSA-13, AuP NPs@CSA-44, and AuP NPs@CSA-131; and AuS NPs@CSA-13, AuS NPs@CSA-44, and AuS NPs@CSA-131) in the majority of cases were higher (MIC values ranging from 0.4 to 3.2 µg/mL) compared with those determined for ceragenins alone (i.e., CSA-13, CSA-44, and CSA-131, MICs ranging from 0.8 to 6.4 µg/mL). Moreover, ceragenin-based nanosystems with CSA-13 and CSA-131 were more effective than those with CSA-44 (Figure 2B).

### 3.3. Serial Passaging Experiment to Induce Development of Resistance towards AuR NPs@CSA-13, AuR NPs@CSA-44, and AuR NPs@CSA-131 by Candida Cells

The initial MICs of AuR NPs@CSA-13, AuR NPs@CSA-44, and AuR NPs@CSA-131 for *C. albicans* ATCC 26790 were 1.6 μg/mL, 3.2 μg/mL, and 1.6 μg/mL, respectively. After two serial passages, the MIC of AuR NPs@CSA-13 increased two-fold, and after 13 serial passages it was enhanced four-fold. It is noteworthy that this value remained to the end of the experiment (Figure 3A). In turn, in relation to AuR NPs@CSA-44 after four serial passages, the MIC of increased two-fold, and no further changes were observed through the rest of the experiment (Figure 3C). On the other hand, the MIC of AuR NPs@CSA-131 was unchanged through 23 passages, and after this passage, the MIC grew two-fold through the end of the investigation (Figure 3E).

With regard to *C. krusei* 156, the initial MICs of AuR NPs@CSA-13, AuR NPs@CSA-44, and AuR NPs@CSA-131 were 0.8 μg/mL, 1.6 μg/mL, and 0.8 μg/mL, respectively. Notably, no change of MIC of AuR NPs@CSA-13 was observed until the end of the experiment (Figure 3B). In the case of AuR NPs@CSA-44 and AuR NPs@CSA-131, the MIC increased two-fold after 21 or 22 passages, respectively (Figure 3D,F). Moreover, no substantial changes through the rest of the experiment were observed.

### 3.4. Antifungal Activity of Nanosystems Involves Generation of Reactive Oxygen Species, Depolarization, and Disruption of the Cell Membrane as Well as Protein Leakage

Reactive oxygen species (ROS) are synthesized in various metabolic pathways during the reduction and oxidation, and constitute a mechanism of action of some antifungal agents [58]. For the assessment of the generation of the ROS by *C. albicans* 1408 cells in response to ceragenins and ceragenins conjugated with NPs, the DCFH-DA assay was performed (Figure 4A–C). The application of a 10 µg/mL concentration of the antifungal agents resulted in a 1.31 to 2.52-fold increase of ROS generation in comparison with the 5 mM hydrogen peroxide. Significantly, this effect was observed for all of the developed nanosystems. On the other hand, free ceragenins, i.e., CSA-13, CSA-44, and CSA-131, enhanced ROS generation from 1.22 to 1.7-fold in relation to the 5 mM hydrogen peroxide.

As shown in Figure 5, no fluorescent signal was detected after the incubation of *C. albicans* 1408 without the test agents, whereas a strong signal from the DFCH-DA fluorescent probe was observed under the treatment with increasing concentrations of nanoparticles.

Activities of the ceragenins and gold nanosystems with ceragenins to cytoplasmic membrane depolarization were determined using the membrane potential-sensitive dye, i.e., diSC_(3)_. The highest increase of fluorescence signals was detected for peanut-shaped Au NPs functionalized with CSA-13, and CSA-131. It should be highlighted that treatment with nanosystems at a concentration of 10 μg/mL resulted in an approximately 1.27 to 2.30-fold increase of the fluorescence intensity compared with a 1 μg/mL concentration of the amphotericin B (Figure 6A–C). In contrast, the same dose of CSA-13, CSA-44, and CSA-131, caused only 1.22 to 1.41-fold increases of fluorescence as compared to 1 μg/mL concentration of the amphotericin B.

The ability of ceragenin-based nanosystems and ceragenins to disrupt the outer layer of *C. albicans* 1408 was determined using the NPN uptake assay (Figure 7A–C). The highest level of membrane permeabilization was observed for rod-shaped Au NPs. In general, the antifungal activities of nanosystems were higher than those observed for CSA-13, CSA-44, and CSA-131 in a free form. For instance, under treatment with AuR NPs@CSA-44 at a concentration of 10 μg/mL, an approximately 2.18-fold (Figure 7B) increase of intensity of the fluorescence signal was recorded compared to 1 μg/mL concentration of the amphotericin B.

To verify whether the treatment of *C. albicans* with ceragenins and ceragenins attachment to Au NPs leads to the release of the protein content, a protein leakage assay was performed. The results shown in Figure 8 (panels A–C) indicate a significant release of cytoplasmic proteins as a consequence of the damage to the fungal membrane. The results obtained were reported for all of the synthesized nanosystems in a dose-dependent manner. It is worth noting that in case of AuP NPs@CSA-131 the greatest leakage of intracellular content was detected with 2.51-fold increase in comparison to the 1 μg/mL concentration of the amphotericin B. With ceragenins, substantially lower absorbance values were determined in relation to the nanosystems.

### 3.5. Ceragenin-Based Nanosystems Exhibit Biocompatibility at Fungicidal Doses

To assess the potential toxicity of the tested nanosystems, a haemolysis assay was performed at concentrations corresponding to their range of candidacidal activity. As demonstrated in Figure 9 (panels A–I), the nanosystems did not induce significant haemolysis at doses from 1 to 10 μg/mL, even if the incubation was extended until 12 h. After 1 h of incubation at concentrations ranging from 1 to 10 μg/mL, haemolysis was not higher than 3.5%. Moreover, only 1.68–9.60% of erythrocytes were injured after 6 h of exposure to the nanosystems, and merely 2.23–9.78% after incubation for 12 h.

## 4. Discussion

Despite significant efforts to develop new antimicrobial agents, escalating resistance is still a global public health concern [59,60]. It is noteworthy that *Candida* spp. are the most common opportunistic fungal pathogens [61,62]. Currently, it is assumed that *Candida* spp. colonize the mucosal surfaces in approximately 50–70% of healthy humans. However, it should be underlined that if breaches in the gastrointestinal and cutaneous barriers appear, for example, after gastrointestinal surgery or venous catheters, the commensal fungi can translocate and invade the bloodstream causing candidaemia [5]. It is important to note that fungi interact with the host’s immune system and then exit the intravascular spaces to invade deep tissues of target organs such as the liver, spleen, kidneys, heart, and brain [63]. The characteristics of the fungi involved in this pathogenicity include the adhesion of tissues, the ability to exhibit yeast−hyphal dimorphism, the hydrophobicity of the cell surface, and finally secretion of proteinases and phenotypic switching [14,64,65].

The studies presented herein demonstrate the potent antifungal activity of rod-, peanut-, and star-shaped Au NPs functionalized by CSA-13, CSA-44, and CSA-131 at low concentrations. Importantly, MICs did not exceed 3.2 µg/mL for both referential and clinical strains, which points out the considerable fungicidal activity of ceragenin-based nanosystems (Figure 2). The lowest MIC values were observed for Au NPs functionalized by CSA-13, where in more than half of the cases this value was 0.8 μg/mL (Figure 2).

It is assumed that electrostatic interactions between positively charged gold nanoparticles and the negative charges on the surface of the fungus play an important role [66]. The positive charge of nanosystems results from the (i) method of synthesis, which involves the use of CTAB, and (ii) the positive charge of ceragenins [42]. On the other hand, the chemical groups responsible for the negative charge on the fungal surface most likely contain acidic amino acids and charged polysaccharides as, perhaps, mannoproteins [67], where the phosphomannosylation has an essential role. Phosphomannosylation is a modification of cell wall proteins and thus modified mannans provide a negative charge to the fungus wall [68].

Gold nanoparticles have been proposed in the development of new fungicidal agents. The high activity of indolicidin-gold NP conjugates against fluconazole-resistant *Candida* strains was observed (four-fold decrease in MIC value compared with free indolicidin) [69,70]. Khan et al. also demonstrated that Au NPs are able to increase the activity of methylene blue in combination with nanosystems against *C. albicans* [71]. Likewise, another study showed that free triangular gold nanoparticles and all conjugated triangular gold nanoparticles demonstrate a significant antifungal activity against clinical *C. albicans* strains in contrast to free peptide ligands [72]. According to above mentioned information, Au NPs seems to be a very good carrier for antimicrobial agents like ceragenins used in this study.

The size and shape of metallic nanoparticles have an influence on their chemical, optical, and thermal properties [73]. Wani et al. proved that smaller sized gold nanodiscs have superior fungicidal effects than larger sized gold polyhedral nanocrystals against *C. albicans* [74]. On the other hand, Jebali et al. confirmed that the antifungal properties of silver and gold nanocubes against *Candida* isolates were greater than gold and silver nanospheres and wires [41]. The above assumption was further supported by other results where a potent bactericidal activity of non-spherical nanosystems in comparison to spherical-nanoparticles was observed [39].

Growing resistance to antimicrobials constitute one of the principal public health problems in the 21st century [75]. Therefore, gold nanoparticles appear to be good candidates for the development of new candidacidal agents, not only due to their intrinsic activity, but also in the view of the infrequent development of resistance in pathogens, including fungi [76]. It should be highlighted that the results obtained in this study induced no emergence of significant resistance in relation to fungi against gold nanoparticles. The motivation for selecting rod-shaped nanoparticles to the induction of the development of resistance of *Candida* strains was dictated by the previous research. The results obtained by Piktel et al. showed no substantial differences between rod-, peanut-, and star-shaped gold nanoparticles [39]. Comparable conclusions were also drawn in our other research, where the negligible distinction on viability of *Staphylococcus aureus* Xen30, *S. epidermidis* 175, *Klebsiella pneumoniae* 700603, *K. oxytoca* 329, *Pseudomonas aeruginosa* LESB58, and *P. aeruginosa* 510 among rod-, peanut-, and star-shaped gold nanosystems was observed. It should be emphasized that all of the mentioned nanoparticles were characterized by similar bactericidal effectiveness [37]. In view of the above, we decided to examine only one type of shape. We considered that the outcomes obtained for the rod-shape will be analogue and comparable to star- and peanut-shaped nanosystems.

Importantly, our series of experiments demonstrate no increase or only a two-fold raise in the MICs of nanosystems with *C. krusei* 156 and only two- or four- fold increases with *C. albicans* 26790 after 25 passages (Figure 3A–F). Similarly, Xie et al. [77] and Zheng et al. [78] obtained no resistance development of gold nanoparticles against *S. aureus*. However, Elbehiry et al. [79] showed resistance to the 10-nm Ag NPs (*n* = 4) and to 20-mm Ag NPs (*n* = 10) after 10 passages utilizing 10-nm and 20-nm silver and gold nanoparticles. In contrast, only two strains developed resistance to the 10-nm Au NPs, and three strains to the 20-nm Au NPs.

The wide range of biomedical applications of gold nanosystems prompted the determination of the mechanism of action. Nanoparticles enter *Candida* cells, and release metal ions from their surface, which inhibit the activity or cause the cell damage and finally lead to the cell death [80]. An important role in the mechanism of the fungicidal action of gold nanoparticles is played by ROS. It is estimated that due to Au NP membrane incorporation and penetration ability, interaction with fungal mitochondria leads to membrane disruption, which interferes with the reduction of molecular oxygen during ATP synthesis. Some amount of oxygen is not reduced causing the formation of superoxide anion and other ROS. This results in a chain reaction involving protein oxidation and oxidation of fatty acid double bonds in the cell membrane, which drastically reduces the fungal membrane integrity. Additionally, ROS are involved in DNA-strand breaks resulting in the loss of cell metabolic function [81,82,83,84,85]. To assess the mechanism of action of ceragenin nanosystems, examination of the ROS generation (Figure 4A–C and Figure 5), diSC_(3)_ assay (Figure 6A–C), NPN assay (Figure 7A–C), and the protein leakage assay (Figure 8A–C) were performed. The results obtained suggest that the principal mechanism of fungicidal action of ceragenin-based nanosystems involves the production of ROS, causing destruction of the fungal membrane and leading to the leakage of intracellular constituents, which ultimately may affect the fungal death. The numbers of reports related to the generation of ROS by Au NPs against fungal strains are still limited, although there is compelling evidence indicating the crucial role of the ROS generation by gold nanoparticles in the antibacterial activity. Mohamed et al. described the antimicrobial effects of Au NPs against *Corynebacterium pseudotuberculosis* via the ROS generation [86]. Furthermore, the ROS-dependent mechanism that ultimately leads to the eradication of pathogens like MDR *E. coli* and methicillin-resistant *S. aureus* was observed by Xie et al. [77] or in the case of ESKAPE strains (*Enterococcus faecium*, *S. aureus*, *K. pneumoniae*, *Acinetobacter baumannii*, *P. aeruginosa*, and *Enterobacter* spp.) by Zheng et al. [78]. According to the protein leakage, Piktel et al. investigated that the cell death upon AuR NP treatment resulted from the loss of membrane integrity, the destabilization, and finally the release of intracellular content from the fungus cells [38].

Knowledge about the potential toxicity and health impact of gold nanoparticles is essential before nanomaterials can be used in clinical settings [87]. In our investigation, an assessment of the haemolytic activity of Au NPs at doses corresponding to MICs and to the fungicidal range was performed (Figure 9A–I). It should be highlighted that no significant differences between the haemolytic activity and the shape of the gold core of nanosystems were observed. Moreover, “candidacidal doses” of Au NPs did not cause significant damage of human RBCs (Figure 9A–I). The results obtained by Rahimi et al. are compatible with the outcomes presented in this research [69]; that is, the hemolytic activity of gold nanoparticles at fungicidal doses is not significant. It is worth noting that the results from the cytotoxic evaluation performed by Raja et al. [88] indicate much higher haemolytic activity of silver nanoparticles (Ag NPs) after 12 h of incubation; haemolysis under treatment with Ag NPs at concentrations of 5 μg/mL, 10 μg/mL, and 25 μg/mL reached 6%, 12.5%, and 85%, respectively. However, in comparison to our investigation, only 1.76%, 9.78%, and 68.99% of the erythrocytes were damaged upon exposure to the same doses of Au NPs, which clearly underlines a greater potential for the use of gold nanomaterials considering their low toxicity.

Collectively, our results suggest that ceragenin nanosystems can be potentially used as a novel and an effective antifungals or drug-delivery carriers to enhance the therapy induced by *Candida*-strains. Nevertheless, further studies are required to determine pharmacodynamic and pharmacokinetic characteristics.

## 5. Conclusions

Functionalization of rod-, peanut-, and star- gold-shaped nanoparticles with CSA-13, CSA-44, and CSA-131 increased fungicidal activity in comparison to the free molecules of ceragenins. Moreover, an important finding of this study is the lack of resistance development to these nanosystems by strains of *Candida*. The fungicidal activity of the developed nanosystems is mediated by the generation of ROS that leads to membrane-permeabilization and release of intracellular content. Additionally, the low hemolytic activity of tested nanosystems at fungicidal doses further underline their high potential in development of new methods to combat infections caused by various *Candida*-strains.

## Figures and Tables

**Figure 1 pharmaceutics-13-01940-f001:**
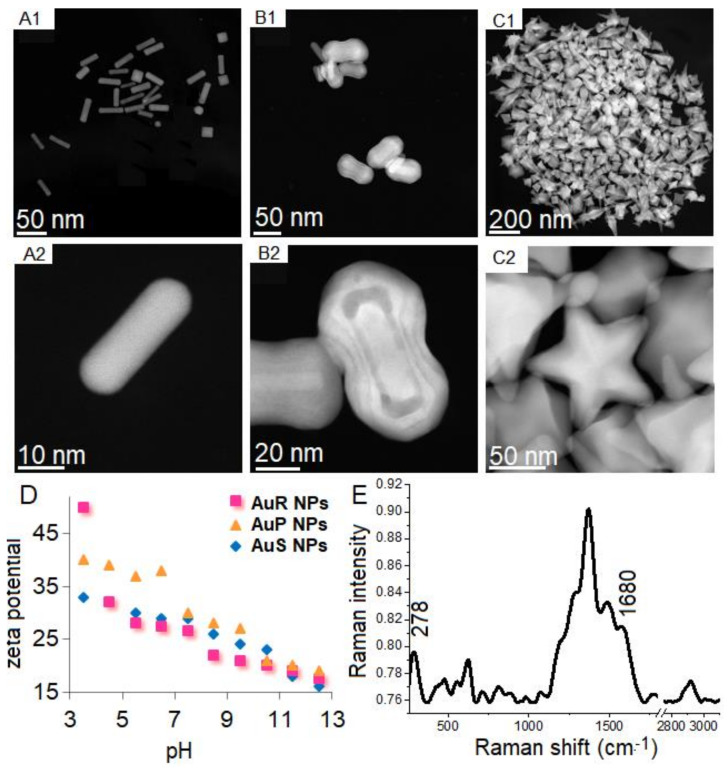
HAADF-STEM over view (1) and magnified (2) images of AuR (**A**), AuP (**B**), and AuS (**C**). Zeta potential values of synthesized AuR, AuP, and AuS (**D**). Representative FT-Raman spectrum showing the chemical mechanism of ceragenins CSA-13 immobilized on the AuP NPs surface (**E**).

**Figure 2 pharmaceutics-13-01940-f002:**
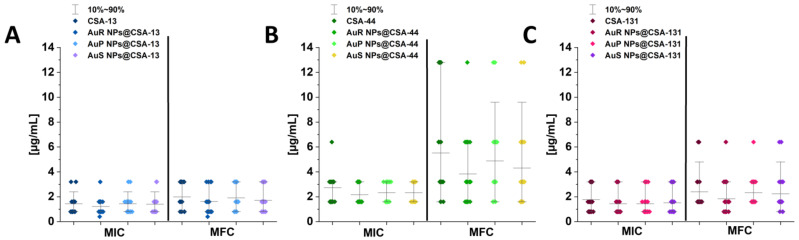
MIC (minimal inhibitory concentration; µg/mL) and MFC (minimal fungicidal concentration; µg/mL) values of CSA-13, AuR NPs@CSA-13, AuP NPs@CSA-13, AuS NPs@CSA-13 (**A**); CSA-44, AuR NPs@CSA-44, AuP NPs@CSA-44, AuS NPs@CSA-44 (**B**); and CSA-131, AuR NPs@CSA-131, AuP NPs@CSA-131, AuS NPs@CSA-131 (**C**) against 19 fungal clinical strains and 2 reference fungal strains.

**Figure 3 pharmaceutics-13-01940-f003:**
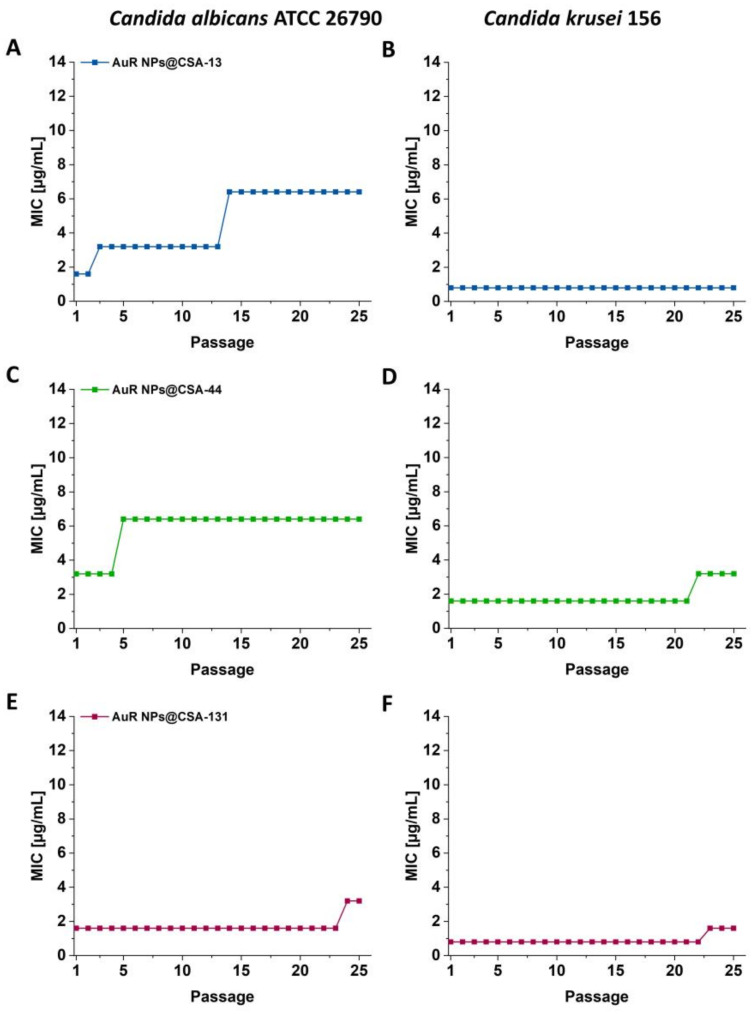
MIC/MFC values for *C. albicans* 26790 (**A**,**C**,**E**) and for *C. krusei* 156 (**B**,**D**,**F**) of the AuR NPs@CSA-13, AuR NPs@CSA-44, and AuR NPs@CSA-131 after 25 serial passages indicated.

**Figure 4 pharmaceutics-13-01940-f004:**
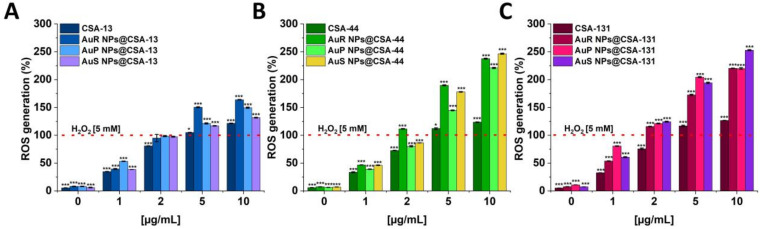
Induction of reactive oxygen species (ROS) generation in *C. albicans* 1408 was evaluated by the DFCH-DA fluorimetric assay. Formation of ROS in cells subjected to CSA-13, AuR NPs@CSA-13, AuP NPs@CSA-13, AuS NPs@CSA-13 (**A**); CSA-44, AuR NPs@CSA-44, AuP NPs@CSA-44, AuS NPs@CSA-44 (**B**); and CSA-131, AuR NPs@CSA-131, AuP NPs@CSA-131, AuS NPs@CSA-131 (**C**) ranging from 1 to 10 μg/mL was presented. The dashed line corresponds to the effect obtained with the 5 mM hydrogen peroxide. The results show the mean ± SD, *n* = 3; * indicates statistical significance at ≤0.05 and *** ≤0.001.

**Figure 5 pharmaceutics-13-01940-f005:**
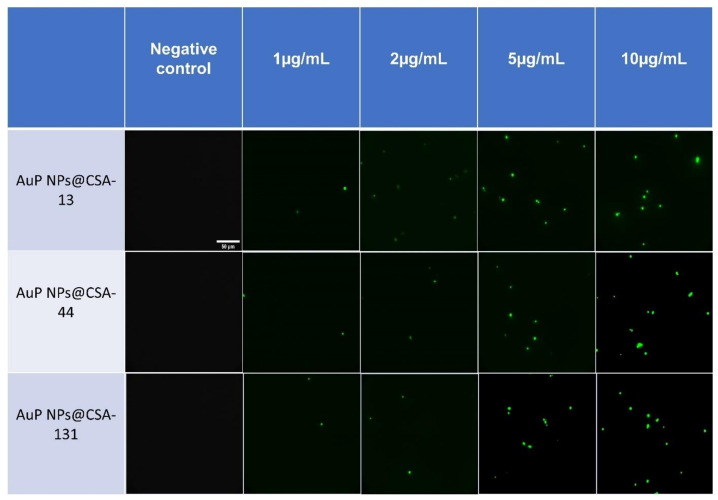
Generation of reactive oxygen species in AuP NPs@CSA-13, AuP NPs@CSA-44, and AuP NPs@CSA-131 -treated *Candida* cells. Increase in dichlorofluorescin diacetate–derived fluorescence signal after 1 h of treatment of *C. albicans* 1408 isolate with tested compounds in concentrations ranging from 1 to 10 μg/mL.

**Figure 6 pharmaceutics-13-01940-f006:**
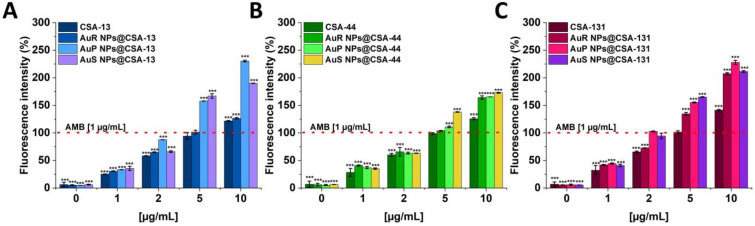
Depolarization of the fungal membrane of *C. albicans* 1408 cell fungal membrane was assessed using diSC_(3)_ assay. The evaluation of the degree of cell membrane depolarization in the presence of CSA-13, AuR NPs@CSA-13, AuP NPs@CSA-13, AuS NPs@CSA-13 (**A**); CSA-44, AuR NPs@CSA-44, AuP NPs@CSA-44, AuS NPs@CSA-44 (**B**); and CSA-131, AuR NPs@CSA-131, AuP NPs@CSA-131, AuS NPs@CSA-131 (**C**) ranging from 1 to 10 μg/mL was evaluated by monitoring the enhancement of intensity of the fluorescence. The dashed line represents the effect obtained with the amphotericin B at a concentration of 1 μg/mL. Results show the mean ± SD, *n* = 3; *** indicates statistical significance at ≤0.001.

**Figure 7 pharmaceutics-13-01940-f007:**
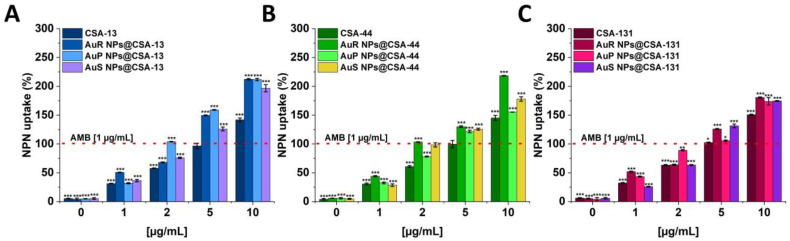
The increase of the membrane permeability of *C. albicans* 1408 subjected to CSA-13, AuR NPs@CSA-13, AuP NPs@CSA-13, AuS NPs@CSA-13 (**A**); CSA-44, AuR NPs@CSA-44, AuP NPs@CSA-44, AuS NPs@CSA-44 (**B**); and CSA-131, AuR NPs@CSA-131, AuP NPs@CSA-131, AuS NPs@CSA-131 (**C**) at doses of 1–10 μg/mL was investigated using the fluorimetric method. The dashed line corresponds to the effect obtained with the amphotericin B at a concentration of 1 μg/mL. The results show the mean ± SD, *n* = 3; * indicates statistical significance at ≤0.05, ** ≤0.01, and *** ≤0.001.

**Figure 8 pharmaceutics-13-01940-f008:**
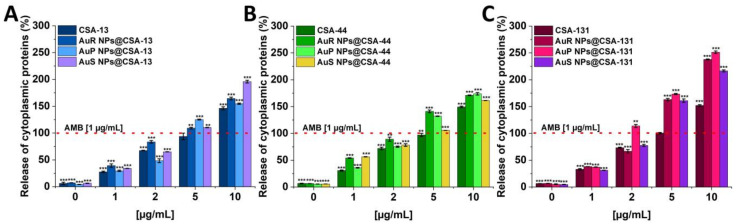
The release of cytoplasmic proteins from *C. albicans* 1408 after treatment with CSA-13, AuR NPs@CSA-13, AuP NPs@CSA-13, AuS NPs@CSA-13 (**A**); CSA-44, AuR NPs@CSA-44, AuP NPs@CSA-44, AuS NPs@CSA-44 (**B**); and CSA-131, AuR NPs@CSA-131, AuP NPs@CSA-131, AuS NPs@CSA-131 (**C**) at concentrations of 1–10 μg/mL was evaluated by the Bradford protein assay. The dashed line represents the effect obtained with the amphotericin B at a concentration of 1 μg/mL. The results show the mean ± SD, *n* = 3; ** indicates statistical significance at ≤0.01, and *** ≤0.001.

**Figure 9 pharmaceutics-13-01940-f009:**
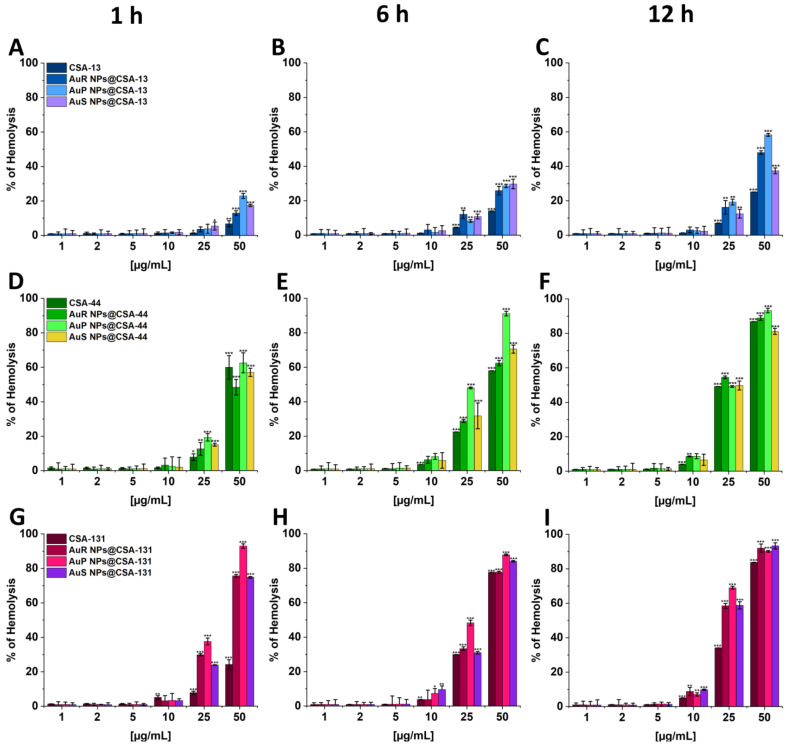
Hemoglobin release from human red blood cells (RBCs) incubated in the presence of CSA-13; AuR-NPs, AuP-NPs, and AuS-NPs containing CSA-13 (**A**–**C**); CSA-44; AuR-NPs, AuP-NPs, and AuS-NPs containing CSA-44 (**D**–**F**); and CSA-131; AuR-NPs, AuP-NPs, and AuS-NPs containing CSA-131 (**G**–**I**) at doses of 1–50 μg/mL after 1 h (**A**,**D**,**G**), 6 h (**B**,**E**,**H**), and 12 h (**C**,**F**,**I**) of incubation. Results show the mean ± SD, *n* = 3; * indicates statistical significance at ≤0.05, ** ≤0.01, and *** ≤0.001.

**Table 1 pharmaceutics-13-01940-t001:** Susceptibility of tested *Candida* isolates to conventional antimitotic agents. S—susceptible; R—resistance; I—intermediate; AMB—Amphotericin B; FLU—Fluconazole; VOR—Voriconazole; CSF–Caspofungin; MCF–Micafungin; FC–Flucytosine; “-“—no breakpoints.

Fungal Strain	Antimycotic Susceptibility ([μg/mL]/Interpretation)					
	AMB	FLU	VOR	CSF	MCF	FC
*C. albicans* 138	0.25/S	0.5/S	0.12/S	0.12/S	0.06/S	1/S
*C. albicans* 166	0.5/S	0.5/S	0.12/S	0.12/S	0.06/S	1/S
*C. albicans* 177	1/S	0.5/S	0.12/S	0.12/S	0.06/S	1/S
*C. albicans* 185	0.25/S	0.5/S	0.12/S	0.12/S	0.06/S	1/S
*C. albicans* 197	0.5/S	0.5/S	0.12/S	0.12/S	0.06/S	1/S
*C. albicans* 1408	1/S	1/S	0.12/S	0.12/S	0.06/S	1/S
*C. albicans* ATCC 26790	1/S	0.5/S	0.12/S	0.12/S	0.06/S	1/S
*C. glabrata* 132	1/S	-/R	0.25/I	0.25/I	0.06/S	1/S
*C. glabrata* 145	1/S	-/R	0.25/I	0.25/I	0.06/S	1/S
*C. glabrata* 154	0.5/S	-/R	0.12/S	0.25/I	0.06/S	1/S
*C. krusei* 137	0.5/S	-/R	0.12/S	0.5/I	0.12/S	16/I
*C. krusei* 148	0.5/S	-/R	0.12/S	0.5/S	0.12/S	16/I
*C. krusei* 156	0.5/S	-R	0.12/S	0.5/I	0.12/S	16/I
*C. krusei* 163	0.5/S	-/R	0.12/S	0.5/I	0.12/S	16/I
*C. krusei* 176	0.5/S	-/R	0.12/S	0.5/I	0.12/S	16/I
*C. krusei* 184	0.5/S	-/R	0.12/S	0.5/I	0.12/S	16/I
*C. tropicalis* 133	0.25/S	0.5/S	0.12/S	0.12/S	0.06/S	1/S
*C. tropicalis* 157	0.5/S	1/S	0.12/S	0.12/S	0.06/S	1/S
*C. tropicalis* 165	0.25/S	1/S	0.12/S	0.12/S	0.06/S	1/S
*C. tropicalis* 178	0.5/S	0.5/S	0.12/S	0.12/S	0.06/S	1/S
*C. tropicalis* 191	0.25/S	0.5/S	0.12/S	0.12/S	0.06/S	1/S

## Data Availability

The data that support the findings of this study are available from the corresponding author upon reasonable request.

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
