# Peer review of "Ceragenin-Coated Non-Spherical Gold Nanoparticles as Novel Candidacidal Agents"

_pharmaceutics, 2021, doi:10.3390/pharmaceutics13111940_

Round 1

Reviewer 1 Report

The manuscript by Sklodowski et al. aims at describing some of the toxic effects (considering the very limited experimental setting this is far from determining a biological mechanism of action, as argued by the authors) of gold NPs as novel anti-Candida agents. The presentation of the study is well performed, but there are significant problems with the experimental setup used that needs to be upgraded in order to attribute meaningfulness to the results. Without that, the results are highly speculative and not necessarily elucidate about the interaction between the different AuNPs and the cells. There are gross mistakes concerning aspects of the physiology and cell biology of Candida species that require correction and there are also points where rigour has to be increased in order to avoid un-necessary generalisations. With proper revision, specially with inclusion of new results that could sustain the findings, the MS could be considered of interest for publication. Specific points follow: 

  1. Concerning the flaws in the experimental setup the main issues with all the performed assays is that we don't know if the probes are indeed inside the cells as the measures were only performed in a population of cells in a spectrofluorimeter. The probes are fluorescent so they allow some fluorescence microscopy in order to determine, at least, that the probes are indeed measuring something happening inside the cells. THe second aspect that requires revision, in my opinion, is that the quantifications are relative and we don't know if an increase in x-fold does have a biological meaning. One way to mitigate this would be to have positive controls, in the case of ROS determination, for example, exposing the cells to a given concentration (maybe an equivalent concentration in terms of growth inhibition to the one used by the authors of their compounds) of H2O2. This is, in my view, essential to provide meaningfulness to the results;
  2. The discussion is very long and most of it describes aspects that should be moved to the introduction (such as the applications of AuNPs). There are many repetitions between the discussion and the introduction; It is absolutely wrong the idea conveyed in the discussion that Candida spp are colonizers of the gastric mucosa and that systemic infections protrude from that. Although this idea is becoming recognized for some species it is for a few of them (C. albicans in most), never Candida spp, as the authors mention. Not all species are inhabitants of the GI tract. Please revise the entire MS for these problems concerning cell physiology to avoid gross mistakes
  3. The introduction reflects several mis-conceptions and wrongly performed generalizations about cell physiology of Candida. I will detail examples. In the introduction it is mentioned 60% of mortality, but this is not true nor accurate as there are many differences between different Candida species and also very much dependent on patient population. Care should be taken. The increase of morbidity would be much more amenable to use. It is not true that the incidence of non-albicans species has increased specially in HIV patients, maybe this is the population studied in the reference mentioned by the authors but if other sources are consulted this will also be true. Revision of the sentence and of the referencing is advised. 
  4. Lines 69-68: The paragraph starts with the idea that modification of the antibiotics is a major mechanism of resistance to antibiotics, which is not true. In fact, the mechanisms that the authors describe afterwards as being associated to antifungal resistance (which are more or less accurate) are perfectly contextualized also in antibacterial resistance. This needs to be modified (and the phrase has un-necessary repetitions);
  5. Line 87: It would be much better if this sentence has been referenced as what is the scenario specifically with the application of NPs against fungi/Candida which is a topic that has been addressed before in the literature, but that has not been reviewed by the authors; It is also a generalization to say that NPs are less prone to develop resistance or that they are less prone to currently developed resistance mechanisms (and so many of them remain to be characterized...);
  6. There are many described examples of NPs (e.g. those of Ag) that are degraded by Candida, specially C. albicans. Again there is a generalization made in line 96. 
  7. The second portion of the introduction is an oversell of the use of NPs. Surely they have several advantages but it is not accurate to say that they can be utilized to mitigate the spread of MDR strains...Maybe in vitro they show potential to inhibit microbes, but where is the evidence that the use of these NPs therapeutically mitigates the spread of MDR? Another generalization;
  8. If the authors have an access to a spectrophotometer why they chose to determine the MIC (which by the way is usually refered as MIC50, MIC90, depending on the level of inhibition it causes, in comparison with a control condition) visually? Isn't it always better to have a quantitative view of the data?
  9. Have you determined cell viability in the serial passages performed? If only based on OD determination, you may be counting cells that are both dead and alive;
  10. The data concerning the classification of the strains as susceptible and resistant to the currently used antifungals should be shown, at least in sup material;
  11. We don't really know if what we are observing concerning protein leakage does reflect leakage as it may also reflect some sort of interaction with the proteins protruding from the cell wall. Have the auhors performed some microscopy observation of the cells to sustain that?

Author Response

Thank you very much for your analysis and suggestions. After examining the attached comments, we have introduced modifications in the content of our manuscript. We hope that provided corrections meet the requirements of the Reviewer.

Reviewer 2 Report

The manuscript received for evaluation deals with some gold nano-systems used as biocidal agents. The subject is not new and it is well documented in literature. It has 94 references, for some of them being also the DOI number indicated.

The chapter 2.2.2 is not well presented and doesn't provide enough data to be reproduced by other scientists; moreover, as an example, the synthesis of gold nano-seeds is misleading- 0.2 M CTAB means around 50 g, and this cannot be mixed with 5 mL aqueous solution of gold salt. Otherwise, the work provide enough data for publication. Figures 2,4,5,6,7 should be re-arranged horizontally. Other small errors- please write subscripts in chemical formulas.

In conclusion, the manuscript can be improved for publication, and after these relatively minor changes can be accepted for publication.

Author Response

(The authors gave the same response as above.)

Reviewer 3 Report

The publication of the article with major corrections of the one entitled “Ceragenin-coated non-spherical gold nanoparticles as novel 2 candidacidal agents”

The article complements the previous one made by the authors and constitutes an advance in the efficacy of the developed nanosystems.

First: Excessive coincidences in the wording with the article titled “Bactericidal Properties of Rod-, Peanut-, and Star-Shaped Gold Nanoparticles Coated with Ceragenin CSA-131 against Multidrug-Resistant Bacterial Strains” and also with the titled NDM-1 Carbapenemase-Producing Enterobacteriaceae are Highly Susceptible to Ceragenins CSA-13, CSA-44, and CSA-131”

Second: Authors must review the writing and go through an anti-plagiarism program for the article, for example, Turnitín to check the amount of matching text and rewrite it.

Third: Check material and methods very carefully and put the reference number, supplying company and country. of all products used. Also, put the reference of the sections used, company and country.

Fourth The authors must perform a microanalysis, for example by means of energy dispersed spectroscopy (EDS) in TEM, to be able to verify the different elements resulting from the fixation and contrast treatment and the presence of gold belonging to their nanoparticles.

Author Response

(The authors gave the same response as above.)

Round 2

Reviewer 1 Report

The MS has been considerably improved by the authors in this revised version and this is very good. I think the fluorescence microscopy experiments that the authors provided in the response to reviewers file should be added to the MS as it is a valuable evidence (maybe you can show only one concentration in the body of the MS and leave the rest for the supp file). The results obtained with H2O2 (and the remaining positive controls of the subsequent experiment) should also be provided in the bar charts so that the reader can better compare the results obtained under the different conditions.

Author Response

Please find the enclosed revised version of manuscript entitled " Ceragenin-coated non-spherical gold nanoparticles as novel candidacidal agents" (submission no: pharmaceutics-1408984). We would like to thank Reviewers for their evaluation and all helpful suggestions. After analyzing the attached comments, we have implemented appropriate amendments in the content of our manuscript that in our opinion significantly improved its quality. We hope that provided corrections meet the requirements of the reviewer, which will allow for the positive evaluation of the manuscript. In addition, a native English speaker carefully edited the manuscript.

In the revised manuscript, all changes are indicated by yellow highlighting.

Thank you for your consideration of our work.

Reviewer 3 Report

The article has improved a lot in the new version, therefore, the publication is recommended

Best regard

Author Response

(The authors gave the same response as above.)
